# Self-Supervised Learning with an Information Maximization Criterion

**Serdar Ozsoy**[1,2] **Shadi Hamdan**[1,3] **Sercan Ö. Arik**[4] **Deniz Yuret**[1,3] **Alper T. Erdogan**[1,2]

[1]KUIS AI Center, Koc University, Turkey    [2]EEE Department, Koc University, Turkey
[3]CE Department, Koc University, Turkey    [4]Google Cloud AI Research, Sunnyvale, CA
{sozsoy19, shamdan17, dyuret, alperdogan}@ku.edu.tr   soarik@google.com

## Abstract

Self-supervised learning allows AI systems to learn effective representations from large amounts of data using tasks that do not require costly labeling. Mode collapse, i.e., the model producing identical representations for all inputs, is a central problem to many self-supervised learning approaches, making self-supervised tasks, such as matching distorted variants of the inputs, ineffective. In this article, we argue that a straightforward application of information maximization among alternative latent representations of the same input naturally solves the collapse problem and achieves competitive empirical results. We propose a self-supervised learning method, CorInfoMax, that uses a second-order statistics-based mutual information measure that reflects the level of correlation among its arguments. Maximizing this correlative information measure between alternative representations of the same input serves two purposes: (1) it avoids the collapse problem by generating feature vectors with non-degenerate covariances; (2) it establishes relevance among alternative representations by increasing the linear dependence among them. An approximation of the proposed information maximization objective simplifies to a Euclidean distance-based objective function regularized by the log-determinant of the feature covariance matrix. The regularization term acts as a natural barrier against feature space degeneracy. Consequently, beyond avoiding complete output collapse to a single point, the proposed approach also prevents dimensional collapse by encouraging the spread of information across the whole feature space. Numerical experiments demonstrate that CorInfoMax achieves better or competitive performance results relative to the state-of-the-art SSL approaches.

## 1   Introduction

Self-supervised learning (SSL) is an important paradigm with significant impact in artificial intelligence [1–6]. In particular, SSL can significantly reduce the need for expensively-obtained labeled data. Furthermore, the internal representations learned by SSL approaches are more suitable for task generalization in transfer learning applications [7]. Recently, SSL methods have even been demonstrated to achieve comparable performance to their supervised counterparts [8, 9].

The primary approach for SSL is to define a pretext objective based on unlabeled data, whose optimization leads to rich and useful representations for downstream applications. For example, for visual processing, a common way to construct a pretext task is to obtain similar features for two different augmentations of the same input, which exploits the expected invariance of representations to particular transformations of the input. However, the corresponding optimization setting should avoid the (total) "collapse problem," which is described as generating the same feature output for all inputs. Figure 1 (a) illustrates a toy view of the total collapse problem, where all latent vector outputs converge to a single point during training. We can also define its less severe version, which

36th Conference on Neural Information Processing Systems (NeurIPS 2022).

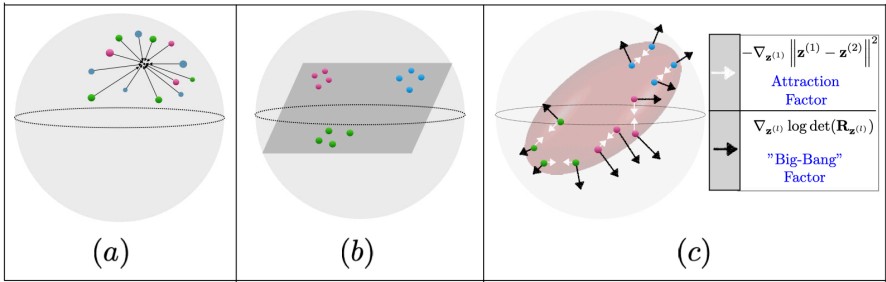

Figure 1: Depictions in latent space - (a) "Total Output Collapse": all latent vectors converge to the same point, (b) "Dimensional Output Collapse": latent vectors are restricted to a strict subspace of the latent space, (c) Gradient dynamics of the CorInfoMax based on (6): the ellipsoid surface reflects the average spread pattern of latent vectors.

is referred to as "dimensional collapse," where all latent vectors lie in a strict subspace of the whole latent space as illustrated by Figure 1.(b).

This article proposes an SSL approach based on the maximization of a form of mutual information among the alternative latent representations obtained from the same input. The mutual information maximization approach would serve two fundamental purposes for SSL:

(i). (*Similarity*) It would enforce the representations obtained for the same input source to be related to or dependent on each other,

(ii). (*No Collapse*) Since the entropy of each alternative representation is lower bounded by the mutual information, maximizing mutual information ensures that latent representations have non-degenerate distributions avoiding collapse.

Therefore, the information maximization approach provides a natural solution to the problems targeted by the existing SSL approaches, which are briefly surveyed in Sec. 2.2.

The conventional choice for mutual information measure is Shannon Mutual Information (SMI) [10]. For SSL, the fundamental limitations on the precise computation of SMI [11–13] impose serious challenges on model training. Especially when mutual information is large, we require drastically more samples (exponentially related to SMI) to obtain a reliable estimate of SMI [13]. However, the use of large batch sizes to improve precision may not be a favorable solution, as (i) it may have an adverse impact on generalization performance [14, 15], (ii) it can significantly increase memory requirements. Furthermore, estimating and optimizing SMI can significantly increase the computational burden. Finally, SMI maximization may induce a nonlinear dependence between the representations corresponding to the same input, where the resultant partitioning of the latent space may not be suitable as an input to a simple linear classifier.

To address these issues, we can consider a second-order-statistics-based mutual information measure, referred to as log-determinant mutual information (LDMI) [16, 17]. The joint entropy measure corresponding to LDMI is the log-determinant of the diagonally perturbed covariance matrix. The zero perturbation case boils down to the Shannon differential entropy for Gaussian distributed vectors. Unlike SMI, which reflects general dependence between its arguments, LDMI reflects linear dependence and is computationally more efficient.

In this article, we propose a novel SSL approach, correlative information maximization (CorInfoMax), derived from the LDMI measure. The loss function for CorInfoMax is obtained through the first-order approximation of LDMI and restricting the linear dependence to the identity map. The resulting optimization objective is a Euclidean distance-based loss function regularized by the log-determinant of the latent vector covariance matrix. This regularization term encourages the spreading of the feature vectors and acts as a natural barrier against the dimensional degeneracy and collapse in the feature space. Our numerical experiments confirm that CorInfoMax learns effective representations that perform well in downstream tasks.

The main contributions can be summarized as follows:

- Based on information theoretical grounds, we propose a novel framework, CorInfoMax, with an interpretable loss function that has explicit components for minimizing variance of positive samples ('the attraction factor') and for making full use of the embedding space to avoid dimensional collapse ('the big-bang factor').
- We introduce a computationally-efficient approximation to LDMI that simplifies it by using an identity mapping instead of a general linear mapping. This approximation directly minimizes representation variance, which is desired for self-supervised learning.
- CorInfoMax relies only on second-order statistics, does not require negative samples, and does not force embeddings to be uncorrelated.
- In addition to these theoretically appealing features, the proposed CorInfoMax framework achieves either better or similar performance results compared to the state-of-the-art SSL approaches.

The following is the organization of the article: Section 2 provides a discussion of the relevant literature related to the proposed approach. We introduce the log-determinant entropy and mutual information measures by highlighting LDMI as a measure of linear dependence in Section 3. In the same section, we derive the objective function of CorInfoMax as a variation on the LDMI measure. Section 4 introduces the proposed correlative information maximization-based self-supervised learning method. Section 5 provides the numerical experiments illustrating the performance of the proposed approach. The appendix provided in the supplementary document contains the details related to these experiments. Finally, Section 6 presents the discussion and conclusions.

## 2 Related work

### 2.1 Determinant maximization for unsupervised learning

The determinant maximization criterion utilized in our framework has been used as an effective algorithmic tool in unsupervised matrix factorization methods such as nonnegative matrix factorization (NMF) [18, 19], simplex-structured matrix factorization (SSMF) [20], sparse component analysis (SCA) [21], bounded component analysis (BCA) [22] and polytopic matrix factorization (PMF) [23].

In the generative models of these frameworks, the input data is assumed to be linear transformations of some latent vectors. Furthermore, these latent vectors are assumed to be sufficiently scattered in their domain. Maximizing the determinant of the latent covariance matrix spreads the latent vector estimates to capture this presumed scattering of the generative model samples. Similarly, the log-determinant of the latent vector covariance matrix in the CorInfoMax objective causes the spreading of latent vectors in their ambient space and, therefore, avoids collapse. According to [17], the determinant maximization criterion used in these matrix factorization frameworks is essentially equivalent to correlative information maximization between input and its latent factors based on the LDMI objective, where the latent vectors are constrained to lie in their domain determined by the generative model.

In the CorInfoMax approach, we utilize determinant maximization for correlative information maximization among latent space vectors rather than between inputs and their latent space representations.

### 2.2 Handling collapse for SSL

Collapse is a central concern for SSL with the same input in computer vision, and we can categorize the existing approaches according to how they handle this issue. For example, contrastive methods, such as SimCLR [8] and MoCo [24], define an objective that pushes features for different inputs (negative samples) away from each other while keeping the representations corresponding to the same input (positive samples) close to each other. The number and the choice of the negative samples appear as critical factors for the performance and scalability of contrastive methods. Another category of SSL is the distillation methods, such as BYOL [4] and SimSiam [25], which avoid collapse through the asymmetry of alternative encoder branches and algorithmic tuning [26]. Other SSL approaches such as DeepCluster [27], SeLa [28] and SvAW [9] enforce a cluster structure in the feature space to prevent constant output. As a different line of algorithms, decorrelated batch normalization (DBN) [29] Barlow Twins [5], Whitening MSE (W-MSE) [30] and VICReg [6], use feature decorrelation [31] as a means to avoid information collapse. The effective whitening mechanisms used in these methods aim for the isotropic spread of information inside the feature space which also prevents "dimensional collapse" illustrated in Figure 1 (b). The reference [32] constructs an SSL loss function using the Hilbert-Schmidt Independence Criterion, a kernel-based independence measure, to maximize the dependence

between alternative representations of the same input. Finally, [33] proposed graph representations for the embeddings of positive samples and the corresponding spectral decomposition algorithm.

The proposed CorInfoMax approach is most related to the decorrelation-based methods discussed above, mainly due to correlation-based measures. However, unlike the decorrelation methods, CorInfoMax does not constrain latent vectors to be uncorrelated. Instead, it avoids covariance matrix degeneracy by using its log-determinant as a regularizer loss function. Furthermore, the information maximization principle is more direct and explicit for the CorInfoMax algorithm.

### 2.3   Information maximization for unsupervised learning

The maximization of SMI has been proposed as an unsupervised learning mechanism in different but related contexts. As one of the earliest approaches, Linsker proposed maximum information transfer from input data to its latent representation and showed that it is equivalent to maximizing the determinant of the output covariance under the Gaussian distribution assumption [34]. Around the same time frame, Becker & Hinton [35] put forward a representation learning approach based on the maximization of (an approximation of) the SMI between the alternative latent vectors obtained from the same image. The most well-known application is the Independent Component Analysis (ICA) Infomax algorithm [36] for separating independent sources from their linear combinations. The ICA-Infomax algorithm targets to maximize the mutual information between mixtures and source estimates while imposing statistical independence among outputs. The Deep Infomax approach [37] extends this idea to unsupervised feature learning by maximizing the mutual information between the input and output while matching a prior distribution for the representations.

It is essential to underline that the proposed method clearly distinguishes itself from the Deep Infomax in [37]: Our objective is not to maximize the mutual information between inputs and outputs of a deep network. Instead, we maximize the mutual information content of the alternative latent representations of the same input. From this point of view, our approach is closer to what is aimed at by [35]. However, we use a different (correlative) information measure, which is computationally more efficient and induces a special form of linear dependence among alternative latent representations of the same input, which may be more desirable considering the goal of generating features for a linear classifier.

## 3   CorInfoMax as a criterion based on log-determinant mutual information

This section derives the optimization objective for the CorInfoMax framework as a variation on the LDMI measure. As described in Appendix A, the LDMI between random vectors $\mathbf{x}$ and $\mathbf{y}$ is given by

$$
\begin{aligned}
I_{LD}^{(\varepsilon)}(\mathbf{x}; \mathbf{y}) = {} & \frac{1}{4}(\log\det(\mathbf{R_x} + \varepsilon\mathbf{I}) + \log\det(\mathbf{R_y} + \varepsilon\mathbf{I}) \\
& - \log\det(\mathbf{R_x} - \mathbf{R_{xy}}(\mathbf{R_y} + \varepsilon\mathbf{I})^{-1}\mathbf{R_{xy}}^T + \varepsilon\mathbf{I}) \\
& - \log\det(\mathbf{R_y} - \mathbf{R_{xy}}^T(\mathbf{R_x} + \varepsilon\mathbf{I})^{-1}\mathbf{R_{xy}} + \varepsilon\mathbf{I})),
\end{aligned}
\tag{1}
$$

where $\mathbf{R_x}$ and $\mathbf{R_y}$ are auto-covariance matrices for $\mathbf{x}$ and $\mathbf{y}$, respectively, and $\mathbf{R_{xy}}$ is their cross-covariance matrix.

There are three potential advantages of using LDMI in (1) over SMI in (A.1) for SSL:

i.  It is based only on second-order statistics. On the other hand, SMI is a statistic based on the joint PDFs of the argument vectors, and its accurate estimation has high sample complexity. In contrast, it is more practical to obtain (the estimates of) the auto-covariance and the cross-covariance matrices, as discussed in Section 4.

ii. There are benefits of information maximization principle in connection with self-supervised training. The maximization of SMI of two vectors induces a general, potentially nonlinear dependence between them, whereas the maximization of LDMI increases correlation, or linear dependence. Therefore, LDMI-based information maximization is expected to organize the feature space being more favorable for data-efficient and low-complexity linear or shallow supervised classifiers as targeted in SSL applications.

iii. The resulting LDMI-based objective functions are more interpretable – they involve the log-determinant of the projector-space covariance, whose maximization clearly avoids feature collapse, a significant concern in SSL methods.

The primary motivation for obtaining an LDMI variant is to increase the similarity between the alternative latent representations of the same input by enforcing the identity transformation between them.

Let $\mathbf{z}^{(1)}$ and $\mathbf{z}^{(2)}$ represent alternative latent representations corresponding to the same input. One potential SSL approach would be to pursue direct maximization of $I_{LD}^{(\varepsilon)}(\mathbf{z}^{(1)}; \mathbf{z}^{(2)})$ in (1). As discussed earlier, this would maximize the correlation between $\mathbf{z}^{(1)}$ and $\mathbf{z}^{(2)}$, therefore inducing a linear dependence between them. For the SSL application, we would prefer the identity mapping to an arbitrary linear relationship such that the alternative latent representations corresponding to the same input concentrate in the same neighborhood of the latent space. Therefore, we modify the LDMI expression in (1) to impose this constraint. For this purpose, we apply the first-order Taylor series approximation, $\log\det(\mathbf{C}+\mathbf{D}) \approx \log\det(\mathbf{C})+\mathrm{Tr}(\mathbf{D}^T\mathbf{C}^{-1})$, on the third term in the right side of (1), which provides

$$
\log\det(\mathbf{R}_{\mathbf{z}^{(1)}} - \mathbf{R}_{\mathbf{z}^{(1)}\mathbf{z}^{(2)}}(\mathbf{R}_{\mathbf{z}^{(2)}} + \varepsilon\mathbf{I})^{-1}\mathbf{R}_{\mathbf{z}^{(1)}\mathbf{z}^{(2)}}^T + \varepsilon\mathbf{I})
$$
$$
\approx \frac{1}{\varepsilon}\mathrm{Tr}(\mathbf{R}_{\mathbf{z}^{(1)}} - \mathbf{R}_{\mathbf{z}^{(1)}\mathbf{z}^{(2)}}\mathbf{R}_{\mathbf{z}^{(2)}}^{-1}\mathbf{R}_{\mathbf{z}^{(1)}\mathbf{z}^{(2)}}^T) + \log\det(\varepsilon\mathbf{I})
$$
$$
= \frac{1}{\varepsilon}\min_{\mathbf{A}_1,\mathbf{b}_1} E(\|\mathbf{z}^{(1)} - (\mathbf{A}_1\mathbf{z}^{(2)} + \mathbf{b}_1)\|_2^2) + P\log(\varepsilon). \tag{2}
$$

Therefore, the expression in (2) corresponds to the mean square error of the best linear (affine) MMSE estimator of $\mathbf{z}^{(1)}$ from $\mathbf{z}^{(2)}$, multiplied by $\varepsilon^{-1}$. Similarly, if we apply the same approximation to the fourth term on the right side of (1), we obtain

$$
\log\det(\mathbf{R}_{\mathbf{z}^{(2)}} - \mathbf{R}_{\mathbf{z}^{(1)}\mathbf{z}^{(2)}}^T(\mathbf{R}_{\mathbf{z}^{(1)}} + \varepsilon\mathbf{I})^{-1}\mathbf{R}_{\mathbf{z}^{(1)}\mathbf{z}^{(2)}} + \varepsilon\mathbf{I})
$$
$$
\approx \frac{1}{\varepsilon}\min_{\mathbf{A}_2,\mathbf{b}_2} E(\|\mathbf{z}^{(2)} - (\mathbf{A}_2\mathbf{z}^{(1)} + \mathbf{b}_2)\|_2^2) + P\log(\varepsilon). \tag{3}
$$

In order to induce the identity mapping between $\mathbf{z}^{(1)}$ and $\mathbf{z}^{(2)}$, we constrain $\mathbf{A}_1 = \mathbf{A}_2 = \mathbf{I}$, and $\mathbf{b}_1 = \mathbf{b}_2 = \mathbf{0}$, which transforms both (2) and (3) into $\varepsilon^{-1}E(\|\mathbf{z}^{(1)} - \mathbf{z}^{(2)}\|_2^2) + \text{const}$. Therefore, scaling the expression in (1) by $4$ and using this modification, we obtain

$$
J(\mathbf{z}^{(1)}, \mathbf{z}^{(2)}) = \log\det(\mathbf{R}_{\mathbf{z}^{(1)}} + \varepsilon\mathbf{I}) + \log\det(\mathbf{R}_{\mathbf{z}^{(2)}} + \varepsilon\mathbf{I}) - 2\varepsilon^{-1}E(\|\mathbf{z}^{(1)} - \mathbf{z}^{(2)}\|_2^2), \tag{4}
$$

where we ignored the constant terms. We refer to (4) as the stochastic CorInfoMax objective function. In Section 4, we propose a SSL method based on the optimization of this objective function.

# 4 LDMI based self-supervised learning

This section introduces the proposed correlative information maximization (CorInfoMax) approach for SSL. We start by describing the presumed setting for the pretext task, which is matching the latent representations of different augmentations of the same input in Section 4.1. Section 4.2 is the main section where we propose the correlative information maximization algorithm for SSL. Finally, we discuss the implementation complexity of the proposed approach in Section 4.3.

## 4.1 Self-supervised learning setting

We start by describing the presumed self-supervised learning setup, which is illustrated in Figure 2:

- *The input* is a sequence of tensors $\{\mathcal{X}[l] \in \mathbb{R}^{M \times K \times N}, l \in \mathbb{Z}\}$, where each index sample corresponds to a batch of $N$ images with dimensions $M \times K$. As discussed in Section 1, the pretext task that we consider is matching the latent representations of the two augmentations of the same input. Therefore, we first apply *the augmentation functions* $a_1(\cdot)$ and $a_2(\cdot)$ to obtain the transformed versions of the input sequence, i.e., $\{\mathcal{X}^{(q)}[l] = a_q(\mathcal{X}[l]) \in \mathbb{R}^{M \times K \times N}, l \in \mathbb{Z}\}, \quad q \in \{1, 2\}$. We will illustrate particular choices of augmentations in the examples provided in Section 5. We note that the proposed setting can be potentially extended to multi-modal schemes where the augmentations are defined for multiple modalities of the same input, such as visual and sound.

- We represent *Siamese networks* to be trained by SSL with the mappings $f_q(\mathcal{U}; \mathbb{W}_{DNN}^{(q)})$ for $q = 1, 2$, where $\mathcal{U} \in \mathbb{R}^{M \times K \times N}$ is the input, $\mathbb{W}_{DNN}^{(q)}$ represents the trainable parameters of the $q^{\text{th}}$ network.

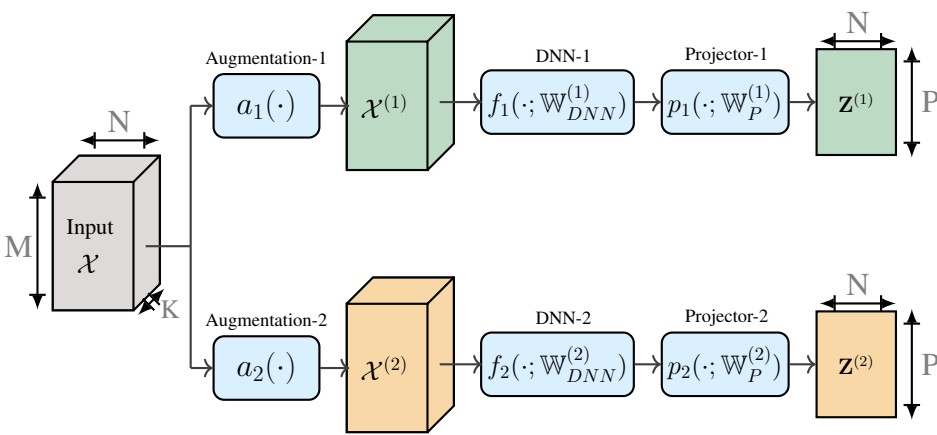

Figure 2: SSL setup: we consider two parallel encoder branches corresponding to two different augmentations of the same input $X$. Augmented views are fed into Siamese networks $f$ followed by projector $p$, basically a 3-layer MLP. $N$ stands for batch size, $M \times K$ for image dimension, $P$ for feature dimension of projector output.

- The outputs of the DNN for different augmented inputs are represented by $\mathbf{Y}^{(q)}[l] = f_q(\mathbb{X}^{(q)}[l]; \mathbb{W}_{DNN}^{(q)}) \in \mathbb{R}^{F \times N}, \quad q \in \{1, 2\}, l \in \mathbb{Z}$, where $F$ is the output feature dimension.
- To be used in self-supervised training, we project the outputs of the enconders using *projector networks* $p_q(\mathbf{Y}; \mathbb{W}_P^{(q)})$, $q = 1, 2$, where $\mathbf{Y}$ is its input, and $\mathbb{W}_P^{(q)}$ is the set of trainable parameters of the $q^{\text{th}}$ projector.
- The outputs of the projector network for different augmentations are represented by $\mathbf{Z}^{(q)}[l] = p_q(\mathbf{Y}^{(q)}[l]; \mathbb{W}_P^{(q)}) \in \mathbb{R}^{P \times N}, \quad q \in \{1, 2\}, l \in \mathbb{Z}$, where $P$ is the projector head dimension.

For the numerical experiments in the article, we consider the weight-sharing setup where both encoders use the same network weights. We provide more details in Appendix B.

### 4.2 Correlative mutual information maximization

We define the CorInfoMax approach for SSL through the optimization problem

$$\underset{\mathbb{W}_{DNN}, \mathbb{W}_P}{\text{maximize}} \quad \hat{J}(\mathbf{z}^{(1)}; \mathbf{z}^{(2)})[l] \tag{5a}$$

where $\hat{J}^{(\varepsilon)}$ is the sample based estimate of (4) at the $l^{\text{th}}$-batch which can be written as

$$\hat{J}(\mathbf{z}^{(1)}, \mathbf{z}^{(2)})[l] = \log\det(\hat{\mathbf{R}}_{\mathbf{z}^{(1)}}[l] + \varepsilon\mathbf{I}) + \log\det(\hat{\mathbf{R}}_{\mathbf{z}^{(2)}}[l] + \varepsilon\mathbf{I}) - \frac{2}{\varepsilon N}\|\mathbf{Z}^{(1)}[l] - \mathbf{Z}^{(2)}[l]\|_F^2, \tag{6}$$

where the rightmost term stands for the sample based estimate of the term $2\varepsilon^{-1}E(\|\mathbf{z}^{(1)} - \mathbf{z}^{(2)}\|_2^2)$ in (4), and $\hat{\mathbf{R}}_{\mathbf{z}^{(1)}}[l]$ and $\hat{\mathbf{R}}_{\mathbf{z}^{(2)}}[l]$ are the auto-covariance matrix estimates for the projector heads at the $l^{\text{th}}$ batch. If the batch size is large enough, these estimates can only be obtained from the current batch samples. However, due to hardware limitations and considering the fact that intermediate batch sizes offer better accuracy [14, 15], sufficiently-large batch sizes may not be possible to obtain reliable covariance estimates. Therefore, we adopt recursive covariance matrix estimation across batches [38]. The corresponding covariance update expressions take the form

$$\hat{\mathbf{R}}_{\mathbf{z}^{(q)}}[l] = \lambda\hat{\mathbf{R}}_{\mathbf{z}^{(q)}}[l-1] + (1-\lambda)\frac{1}{N}\tilde{\mathbf{Z}}^{(q)}[l]\tilde{\mathbf{Z}}^{(q)}[l]^T. \, q \in \{1, 2\}, \tag{7}$$

where $0 \le \lambda < 1$ is the forgetting factor, and $\tilde{\mathbf{Z}}^{(q)}[l]$ represents the batch of mean-centralized projector outputs defined as $\tilde{\mathbf{Z}}^{(q)}[l] = \mathbf{Z}^{(q)}[l] - \boldsymbol{\mu}^{(q)}[l]\mathbf{1}_N^T$, where $\{\boldsymbol{\mu}^{(q)}[l], q \in \{1, 2\}\}$ represents the mean estimates for the projector outputs, updated by $\boldsymbol{\mu}^{(q)}[l] = \lambda\boldsymbol{\mu}^{(q)}[l-1] + (1-\lambda)\frac{1}{N}\mathbf{Z}^{(q)}\mathbf{1}_N$, for $q \in \{1, 2\}$. (See Appendix C for CorInfoMax pseudocode.)

It is informative to inspect the terms in the objective function $\hat{J}(\mathbf{z}^{(1)}, \mathbf{z}^{(2)})$ in (6):

- Minimization of $\frac{2\varepsilon^{-1}}{N}\|\mathbf{Z}^{(1)}[l] - \mathbf{Z}^{(2)}[l]\|_F^2$, acts as a force to pull the representations of alternative augmentations toward each other, which we refer to as the *attraction factor*.

- Maximization of $\log\det(\hat{\mathbf{R}}_{\mathbf{z}^{(1)}}[l] + \varepsilon\mathbf{I})$ acts as a dispersion force causing non-degenerate (for $\varepsilon \approx 0$) expansion of projection vectors in the $P$-dimensional space, which we informally refer to as *big-bang factor*. Therefore, the covariance determinant acts as a regularization or barrier function avoiding collapse in the latent space. Consequently, it provides a convincing replacement for negative samples in contrastive methods to prevent feature collapse. Appendix K illustrates the eigenvalues of the projector covariance vector for the CIFAR-10 training, which demonstrates that the CorInfoMax criterion spreads information throughout the feature space, avoiding dimensional collapse.

Figure 1.(c) illustrates the learning dynamics of the CorInfoMax approach based on (6) with a toy picture. In this figure, the ellipsoid is a representative surface for the level set of the quadratic function $(\mathbf{z} - \boldsymbol{\mu}^{(q)})^T \mathbf{R}_{\mathbf{z}^{(q)}}^{-1}(\mathbf{z} - \boldsymbol{\mu}^{(q)})$, which reflects the spreading pattern of the latent vectors around their mean. The black arrows represent the gradient of the $\log\det(\mathbf{R}_{\mathbf{z}^{(q)}})$ regularization factor, which acts as a force to push latent vectors away from the center of the ellipsoid ($\boldsymbol{\mu}^{(q)}$) and therefore corresponds to the expansion force of the "big-bang" factor. In the same figure, the white arrows correspond to the gradients corresponding to the attraction factor, i.e., the Euclidean distance between two positive pairs. In summary, we can view the learning dynamics of CorInfomax SSL as an expansion in the latent space, while the representations of positive samples are attracted to each other. More information on the CorrInfoMax optimization setting is provided in the Appendix M. Furthermore, Appendix J contains a visualization of the embeddings obtained using the CorInfoMax criterion.

### 4.3 Computational complexity

The main difference between our and related SSL methods is the extra log-determinant computation. The log determinant is typically computed using a decomposition method such as the LU decomposition, whose gradient requires a matrix inversion. Both determinant and inversion have the same complexity as matrix multiplication [39], i.e. $O(n^\alpha)$ with $2 < \alpha \leq 3$ for multiplying two $n \times n$ matrices. In our experiments with GPUs, we observe an almost flat runtime cost up to $n = 1024$, which implies that the overhead due to the $\log\det$ cost is insignificant in practice. We find the extra cost of log determinant and its gradient to be negligible compared to the rest of the model computation, whose runtime is dominated by the encoder, as confirmed by the runtime experiments reported in Appendix G.

## 5 Experiments

### 5.1 Implementation details

*Datasets:* We perform experiments on CIFAR-10, CIFAR-100 [40], Tiny ImageNet [41], COCO [42], ImageNet-100 and ImageNet-1K [43] datasets[1]. For ImageNet-100, we use the same subset of ImageNet-1K [43] as related work [44–46]. (See Appendix D for more details.)

*Training Procedure:* The experiments consist of two consecutive stages: pretraining and linear evaluation. We first perform the unsupervised pretraining of the encoder network $f$ by applying the proposed CorInfoMax method described in Section 4.2 on the training dataset. After completing pretraining, we perform the linear evaluation, a standardized protocol to evaluate the quality of the learned representations [4, 8, 47].

For the linear evaluation stage, we first perform supervised training of the linear classifier using the representations obtained from the encoder network $f$ with frozen coefficients on the same training dataset. Then, we obtain the test accuracy results for the trained linear classifier based on the validation dataset.

---

[1] CorInfoMax's source code is publicly available in `https://github.com/serdarozsoy/corinfomax-ssl`

*Computational Resources:* For ImageNet-100 and ImageNet-1K, we pretrain our model on up to 8 A100 Cloud GPUs. The remaining datasets are trained using a single T4 and V100 Cloud GPU. Linear evaluations are performed using the same type and amount of computational resources. The details related to batch sizes are described in Section 5.1.3.

### 5.1.1 Input augmentations

During the *pretraining stage*, two augmented versions of each input image are generated as shown in Figure 2. During this process, each image is cropped with random size, resized to the original resolution, followed by random applications of horizontal mirroring, color jittering, grayscale conversion, Gaussian blurring, and solarization. Since we use the same augmentation parameters as BYOL [4] and VicReg [6]: each augmentation branch uses the same probability values for these randomized operations except Gaussian blurring and solarization, which use different probabilities.

During the training phase of the *linear evaluation* stage, a single augmentation of each input image is produced by random cropping and resizing followed by a random horizontal flip. For the test phase of linear evaluation, we use resize and center crop augmentations, similar to [5, 6]. We provide more details about the augmentations in Appendix E.

### 5.1.2 Network architecture

*Encoder Network*: For CIFAR datasets, we use a modified form of ResNet-18 architecture [48] similar to [8, 25, 33]. We use standard ResNet-50([48]) for Tiny ImageNet, ImageNet-100 and ImageNet-1K, and also standard ResNet-18([48]) for ImageNet-100. In all cases, the last fully connected layer is removed. Therefore, the encoder output size is 512 and 2048 for ResNet-18 and ResNet-50, respectively. The encoder network $f$ shares weights between augmented branches (see Appendix B).

*Projector Network:* The output of the encoder network is fed into the projector network, as in Figure 2. The projector network $p$ is a 3-layer MLP, with ReLU activation functions for the hidden layers and linear activation functions for the output layer. The projector dimensions are 2048-2048-64 for the CIFAR-10 dataset, 4096-4096-128 for the CIFAR-100, Tiny ImageNet and ImageNet-100, and 8192-8192-512 for the ImageNet-1K. Finally, we perform the $\ell_2$-normalization on the projector output.

*Linear Classifier:* For the linear evaluation phase, we employ a standard linear classifier whose input is the weight-frozen encoder network's output.

### 5.1.3 Optimization

For pretraining, we use 1000 epochs with a batch size of 512 for CIFAR datasets, and 800 epochs with a batch size of 1024 for Tiny ImageNet. For ImageNet-100 experiments, we use 400 epochs for ResNet-18 and 200 epochs for ResNet-50, with a batch size of 1024 for both. ImageNet-1K experiments are conducted as 100 epochs with a batch size of 1536. We use the SGD optimizer with a momentum of 0.9 and a weight decay of $1 \times 10^{-4}$. The initial learning rate is 0.5 for CIFAR datasets and Tiny ImageNet, 1.0 for ImageNet-100, and 0.2 for ImageNet-1K. These learning rates follow the cosine decay with a linear warmup schedule.

We use the modified form of (6) as our objective function, where we replace $\frac{2\varepsilon^{-1}}{N}$ with $\alpha$, the attraction coefficient, which we consider as a separate hyper-parameter. In our experiments, the diagonal perturbation is $\varepsilon = 1 \times 10^{-8}$, while $\alpha = 250$ for CIFAR-10, $\alpha = 1000$ for CIFAR-100, $\alpha = 2000$ for ImageNet-1K, and $\alpha = 500$ for Tiny ImageNet and ImageNet-100. The forgetting factor $\lambda = 0.01$ for all datasets except Tiny ImageNet and ImageNet-1K, which have $\lambda = 0.1$. Details with coefficients of our loss function are provided in Appendix F.

For linear evaluation, the linear classifier is trained for 100 epochs with a batch size of 256 for all datasets. We used the SGD optimizer with a momentum of 0.9, and without weight decay. For all datasets except ImageNet-1K, cosine decay schedule is utilized with initial and minimum learning rates of 0.2 and $2 \times 10^{-3}$ respectively. For ImageNet-1K, we use a step scheduler with a starting value of 25, which is reduced by a factor of 10 every 20 epoch.

## 5.2 Results

We evaluate the learned representations from the CorInfoMax pretraining by following the linear evaluation protocol explained in Sec. 5.1. Table 1 shows that CorInfoMax achieves state-of-the-art performance in linear classification after pretraining. Due to the limited size of the validation sets (5-10K), differences less than $\approx 0.5\%$ are not statistically significant. The full comparison is provided in Table 10 in Appendix H. It is also interesting to observe the progress of the LDMI measure during the CorInfoMax training process, which is illustrated in Appendix L.

Table 1: Top-1 accuracies (%) under linear evaluation on different datasets. Results are reported from [30, 33, 49] for CIFAR-10 and CIFAR-100, [33] for Tiny ImageNet,[46, 50] for ImageNet-100 (IN-100) in in ResNet-50, [49] for ImageNet-100 (IN-100) in ResNet-18, [25, 33] for ImageNet-1K (IN-1K). In the case of the result of a model in more than one resource, we integrate the largest score. We **bold** all top results that are statistically indistinguishable.

| Method | CIFAR-10 | CIFAR-100 | Tiny-IN | IN-100 | | IN-1K |
| | ResNet-18 | ResNet-18 | ResNet-50 | ResNet-18 | ResNet-50 | ResNet-50 |
|---|---|---|---|---|---|---|
| SimCLR [8] | 91.80 | 66.83 | 48.12 | 77.04 | - | 66.5 |
| SimSiam [25] | 91.40 | 66.04 | 46.76 | 78.72 | 81.6 | 68.1 |
| Spectral [33] | 92.07 | 66.18 | 49.86 | - | - | 66.97 |
| BYOL [4] | 92.58 | 70.46 | - | **80.32** | 78.76 | **69.3** |
| W-MSE 2 [30] | 91.55 | 66.10 | - | 69.06 | - | - |
| MoCo-V2 [51] | **92.94** | 69.89 | - | 79.28 | - | 67.4 |
| Barlow [5] | 92.10 | 70.90 | - | **80.38** | - | 68.7 |
| VICReg [6] | 92.07 | 68.54 | - | 79.40 | - | 68.6 |
| CorInfoMax | **93.18** | **71.61** | **54.86** | **80.48** | **82.64** | 69.08 |

We compare the semi-supervised learning performance of CorInfoMax with VICReg [6]. For a fair comparison, VICReg is pretrained for 100 epochs on ImageNet-1K, then semi-supervised learning performances of both models are evaluated by fine-tuning the encoders with $1(\%)$ and $10(\%)$ of the labeled ImageNet-1K dataset. The results are presented in Table 2, while details are provided in Appendix F.5.

Table 2: Top-1 accuracies (%) under semi-supervised classification on ImageNet-1K dataset after 100 epoch pretraining. VICReg is pretrained and evaluated using hyper-parameters reported in [6].

| Method | 1% of samples | 10% of samples |
|---|---|---|
| VICReg | 44.75 | 62.16 |
| CorInfoMax | 44.89 | 64.36 |

We experimented with transfer learning on object detection and instance segmentation with the same procedure and code of MoCo[52], and reproduce the results with the MoCo V2 [53] checkpoint. Our algorithm shows competitive performance with MoCo V2 [53], related details are provided in Appendix I.

## 6 Discussions and conclusions

We proposed a novel SSL framework based on the correlative information maximization, CorInfoMax, which provides a natural solution for obtaining organized representations that are scattered in the latent space and avoiding collapse. Our experiments demonstrate the state-of-the-art performance of CorInfoMax in numerous downstream tasks. The loss function of CorInfoMax does not have any significant impact on training time. Future work can further improve CorInfoMax focusing on shortcomings like improved selection of hyperparameter and augmentation, a core problem in SSL overall.

## Acknowledgments and Disclosure of Funding

This work/research was supported by KUIS AI Center Research Award. We gratefully acknowledge the support of Google Cloud Credit Award.

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
