# OpenReview forum: "Self-Supervised Learning with an Information Maximization Criterion"
_NeurIPS.cc/2022/Conference — NeurIPS 2022 Accept_

### Official Review · Reviewer_GZwK · 2022-07-09

**Rating:** 4
**Confidence:** 5
**Soundness:** 2 fair
**Presentation:** 3 good
**Contribution:** 2 fair

**Summary:**

This paper presents a new self-supervised learning method based on the log-determinant mutual information proposed earlier. Experiments on three small size datasets and one medium size dataset show the effectiveness of the proposed method.

**Questions:**

1. This article only contains the results of classification tasks, and the performance of the proposed method on tasks such as detection, transfer, and semi-supervised learning also needs to be given.
2. The classification results on ImageNet-1000K should be reported.
3. The lines 159-162 are difficult to read. I don't understand that $h_{LD}^*\left( {x\left| y \right.} \right)$ can be treated as a measure of remaining uncertainty. I hope the author can give a detailed explanation.
4. This paper claims that the linear property of LDMI is the key to good performance. However, from " Self-Supervised Learning with Kernel Dependence Maximization", we can know that the nonlinear property of the Kernel Dependence is the key to good performance. I hope the author can give in-depth analysis and comparison.
5. Barlow-Twins is also proposed to explore the linear correlation between different augmentations of the same sample. I hope the author can give the difference and connection between the proposed method and Barlow-Twins, both experimentally and theoretically.
6. For large-scale dataset such as ImageNet-1000, the log-det cost is significant in practice.

**Limitations:**

The authors have adequately addressed the limitations and potential negative societal impact of their work.

**Strengths And Weaknesses:**

Strengths:
1. The experimental results are good.
2. Overall, the article is easy to read.

Weaknesses:
1. The experimental part is relatively weak.
2. The novel of this paper is limited. The main contribution of this paper is to directly generalize the earlier proposed log-determinant mutual information to the field of self-supervised learning.
3. Section 3 is superfluous.

---

> ### Author Response · Authors · 2022-08-02
> **Reponse to Reviewer GZwK Part 4 of 4**
>
> > Barlow-Twins is also proposed to explore the linear correlation between different augmentations of the same sample. I hope the author can give the difference and connection between the proposed method and Barlow-Twins, both experimentally and theoretically.
>
> In relation to this point, in Section 2.2 of our article, we have the following.
>
> "As a different line of algorithms, decorrelated batch normalization (DBN) [28] Barlow-Twins [5], Whitening MSE (W-MSE) [29] and VICReg [6], use feature decorrelation [30] as a means to avoid information collapse. The effective whitening mechanisms used in these methods aim for the isotropic spread of information inside the feature space which also prevents "dimensional collapse"
> illustrated in Figure 1.(b). ...
>
> The proposed CorInfoMax approach is most related to the decorrelation-based methods discussed
> above, mainly due to correlation-based measures. However, unlike decorrelation methods,
> CorInfoMax does not constrain latent vectors to be uncorrelated. Instead, it avoids covariance matrix
> degeneracy by using its log-determinant as a regularizer loss function. Furthermore, the information
> maximization principle is more direct and explicit for the CorInfoMax algorithm}"
>
> To further clarify the relation of the proposed CorInfoMax framework to Barlow Twins:
>
> Barlow Twins approach is based on the idea of redundancy reduction for the representation, where the feature components are forced to be decorrelated. The corresponding loss function is
> $$
> \mathcal{J}\_{BarlowTwins}=\|\mathbf{C}\_{z_1,z_2}-\mathbf{I}\|_F^2
> $$
>
> where $\mathbf{C}\_{z_1,z_2}$ is the correlation coefficient matrix. Minimization of this loss function results in white (decorrelated) feature vectors.
>
> There is a certain resemblance between Barlow Twins and the proposed CorInfoMax approaches, mainly based on the use of the covariance measure.  However, unlike the Barlow Twins approach, the representations are not forced to be uncorrelated in CorInfoMax. In fact, the feature covariance matrices obtained in experiments with different datasets are typically not diagonal.
>
> Furthermore, related to the connection with Information Theory, Appendix C of [5] poses the Barlow Twins loss function as
> an approximation of the Information Bottleneck objective
> $$
> \mathcal{IB}(\theta)=\mathcal{I}(Z_\Theta,Y)-\beta\mathcal{I}(Z_\Theta,X),
> $$
>
> where $X$ and $Y$ are original and distorted inputs, respectively, $Z$ is the output of the encoder for $Y$, and $\mathcal{I}(\cdot,\cdot)$ is the Shannon mutual information. According to this interpretation, the Barlow Twins approach tries to maximize mutual information between input and output, while reducing redundancy at output by eliminating the information content related to distortion.  In contrast, the CorInfoMax approach is based on maximization of the correlative information between the representations of random distortions of the same input.
>
> In terms of experimental results, based on Table 1 in our article, the Top-1 accuracy values are
>
> - For CIFAR-10, $92.10$ for Barlow Twins, $93.18$ for CorInfoMax,
> - For CIFAR-100, $70.90$ for Barlow Twins, $71.61$ for CorInfoMax,
> - For Imagenet-100, $80.38$ for Barlow Twins, $80.48$ for CorInfoMax.
> - For Imagenet-1000 (100 epochs), $68.7$ for Barlow Twins, $69.08$ for CorInfoMax.
>
> Similarly, the Top-5 accuracy values are
>
> - For CIFAR-10, $99.73$ for Barlow Twins, $99.88$ for CorInfoMax,
> - For CIFAR-100, $91.91$ for Barlow Twins, $92.40$ for CorInfoMax,
> - For Imagenet-100, $95.28$ for Barlow Twins, $95.46$ for CorInfoMax.
>
> > For large-scale dataset such as ImageNet-1000, the log-det cost is significant in practice.
>
> Section 4.3 (Computational Complexity) and Appendix G (Algorithm runtime results) address the issue of the impact of $\log\det$ covariance term on computational complexity. Based on your comment, we have further extended this discussion to include run-time results for the Imagenet-1K dataset. These results are reported in Table 9 of Appendix G. From these results, we observe that the impact of the $\log\det$ function on computational complexity is negligible compared to computational complexity due to the other parts of model training. This is mainly due to the fact that the encoder complexity scales proportionally to the input dimension, whereas the complexity of $\log\det$ is dependent only on the projector dimension. In fact, as demonstrated in Table 9 of Appendix G, for the Imagenet-1K dataset and for a projector dimension of $1024$ and for a batch size of $512$, the computation $\log\det$ contributes only $2.83$ percent of the total computation time, which is negligible. This percentage  drops further with increasing batch sizes.

---

> ### Author Response · Authors · 2022-08-02
> **Reponse to Reviewer GZwK Part 3 of 4**
>
> > This paper claims that the linear property of LDMI is the key to good performance. However, from "Self-Supervised Learning with Kernel Dependence Maximization", we can know that the nonlinear property of the Kernel Dependence is the key to good performance. I hope the author can give in-depth analysis and comparison.
>
> This is a really fair point. The reference "Self-Supervised Learning with Kernel Dependence Maximization" is directly relevant to the framework of our article, and therefore we have included in the Related work subsection of the revised article. In summary, the reference claims:
> - The performance of the mutual information maximization-based SSL schemes are dependent on the choice of the surrogate function reflecting mutual information,
> - The the Hilbert-Schmidt Independence Criterion (HSIC) which reflects kernel-based dependence between its arguments is suitable for constructing SSL objectives, mainly due to its connections to existing successful SSL measures, the existence of unbiased estimator for HSIC that requires relatively small sample size, and the empirical evidence provided in the article.
>
> The SSL-HSIC objective in the article is given by
> $$
>      \mathcal{L}\_{SSL-HSIC}(\theta)=-HSIC(Z,Y)+\gamma \sqrt{HSIC(Z,Z)},
> $$
>
>  where $Z$ is an encoder-based representation of the randomly transformed input image, $Y$ is a one-hot encoded representation of the input image. The HSIC measure is estimated using
>  $$
>      \widehat{HSIC}(Z,Y)=\frac{1}{(N-1)^2}\text{Tr}(\mathbf{KHLH}),
> $$
>
> where $\mathbf{K}\_{ij}=k(\mathbf{z}_i,\mathbf{z}_j)$ and $\mathbf{L}\_{ij}=l(\mathbf{y}\_i,\mathbf{y}\_j)$ are kernel matrices corresponding to arguments with kernel functions defined as $k(\mathbf{z},\mathbf{z}')=\langle \phi(\mathbf{z}),\phi(\mathbf{z}') \rangle$ and $l(\mathbf{y},\mathbf{y}')=\langle \psi(\mathbf{y}),\psi(\mathbf{y}') \rangle$ ($\phi(\cdot)$ and $\psi(\cdot)$ are nonlinear transformation functions) and $\mathbf{H}$ is the centering matrix.
>
> In Section 5 of the same article, it is reported that the best empirical performance is achieved by the choice of nonlinear kernels for the $100$-epoch Imagenet-1000 experiment, which is potentially the basis of the claim about superiority of nonlinear kernels.
>
>  We first note that, for the choice of linear kernels, $\mathcal{L}\_{SSL-HSIC}(\theta)$ would be equivalent to
> $$
>  \mathcal{L}\_{SSL-HSIC}(\theta)=-\text{Tr}(\hat{\mathbf{R}}\_{ZY})+\gamma\sqrt{\text{Tr}(\hat{\mathbf{R}}\_{Z})}
> $$
>
> where $\hat{\mathbf{R}}\_{ZY}$ and $\hat{\mathbf{R}}\_{Z}$ are the estimates of the cross-covariance and the auto-covariance matrices of the arguments, respectively, which is not equivalent to the LDMI measure between $Z$ and $Y$ given by
> $$
>  \text{LDMI}(Z,Y)=\frac{1}{2}\log\det(\hat{\mathbf{R}}\_{Z}+\epsilon \mathbf{I})-\frac{1}{2}\log\det(\hat{\mathbf{R}}\_{Z}-\hat{\mathbf{R}}\_{ZY}(\hat{\mathbf{R}}\_{Y}+\epsilon \mathbf{I})^{-1}\hat{\mathbf{R}}\_{YZ}).
> $$
>
> Comparing both objectives, we observe that $\mathcal{L}\_{SSL-HSIC}$ is almost a linear function (except for the square root term) of the cross-covariance and autocovariance matrix estimates of the representations, whereas the LDMI objective is their nonlinear function. We can make the same observation about the CorInfoMax objective derived from the LDMI:
> $$
>      \text{CorInfoMax}(Z,Y)=-\alpha \lVert\mathbf{Z}-\mathbf{Y}\rVert_{F}^2+\log\det(\hat{\mathbf{R}}\_{Z})
> $$
>
> Therefore, the optimization of the CorInfoMax objective functions, derived from the LDMI measure, does not correspond to the loss function of the linear kernel case of the SSL-HSIC. In other words, these separate SSL criteria, i.e., the CorInfoMax and the SSL-HSIC with linear kernel, optimize different nonlinear functions of the correlations of the positive sample embeddings. As a result, the observations about empirical performance of the SSL-HSIC may not directly translate to the CorInfoMax.
>
> In fact, regarding empirical performance corresponding to the 100-epoch Imagenet-1000 experiment, Section 5 of the SSL-HSIC article [32] lists the $65.27$, $66.67$, and $66.72$ top-1 accuracy values for the SSL-HSIC approach with linear, Gaussian, and inverse multi-quadratic kernels, respectively. In our revised article, we have also reported the accuracy level of $69.08$ with the CorInfoMax corresponding to the same setting.
>
>
> In conclusion, the empirical competitive results of the CorInfoMax do not contradict with the findings of the SSL-HSIC article. We note that the performance improvements achieved by CorInfoMax can be attributed to the minimization of representation invariance (for the same input and different distortions) based on the minimization of the error term $\lVert\mathbf{Z}-\mathbf{Y}\rVert_{F}^2$ (the attraction factor) and the spread of representations in the high-dimensional latent space due to the barrier term $\log\det(\hat{\mathbf{R}}\_{Z})$ (the big-bang factor).

---

> ### Author Response · Authors · 2022-08-02
> **Reponse to Reviewer GZwK Part 2 of 4**
>
> > The lines 159-162 are difficult to read. I don't understand that $h_{LD}^*(\mathbf{x}|\mathbf{y})$ can be treated as a measure of remaining uncertainty. I hope the author can give a detailed explanation.
>
> In the revised article, we have moved the background information on the LDMI measure to the Appendix. Moreover, in this new section, we provide more information about the derivation of the conditional LD-entropy  $h_{LD}^{(\varepsilon)}(\mathbf{x}|\mathbf{y})$ and its discussion. Below is the copy of the relevant part on how it is treated:
>
> "we define
> $$h_{LD}^{(\varepsilon)}(\mathbf{x}|\mathbf{y})=\frac{1}{2}\log\det(\mathbf{R}\_\mathbf{x}-\mathbf{R}\_{\mathbf{x}\mathbf{y}}(\mathbf{R}\_\mathbf{y}+\varepsilon\mathbf{I})^{-1}\mathbf{R}\_{\mathbf{x}\mathbf{y}}^T+\varepsilon\mathbf{I})+\frac{r}{2}\log(2\pi e),
> $$
> as the conditional LD-entropy. We note that the argument of the $\log\det$ function above is the auto-covariance of the error for linearly estimating $\mathbf{x}$ from $\mathbf{y}$ with respect to the minimum mean square estimation (MMSE) criterion for $\varepsilon \to 0$ (see, for example, Theorem 3.2.2 of [Kailath 2000]). More precisely, given that $\mu_\mathbf{x}$ and $\mu_\mathbf{y}$ represent the means of $\mathbf{x}$ and $\mathbf{y}$, respectively,  $\hat{\mathbf{x}}\_{MMSE}=\mathbf{R}\_{\mathbf{x}\mathbf{y}}\mathbf{R}\_{\mathbf{y}}^{-1}(\mathbf{y}-\mu_\mathbf{y})+\mu_\mathbf{x}$ is the best linear MMSE estimate of $\mathbf{x}$ from $\mathbf{y}$. Therefore, if we define $\mathbf{e}\_{MMSE}=\mathbf{x}-\hat{\mathbf{x}}\_{MMSE}$, the auto-covariance matrix of $\mathbf{e}\_{MMSE}$ is given by $\mathbf{R}\_{\mathbf{e}\_{MMSE}}=\mathbf{R}\_\mathbf{x}-\mathbf{R}\_{\mathbf{x}\mathbf{y}}\mathbf{R}\_{\mathbf{y}}^{-1}\mathbf{R}\_{\mathbf{x}\mathbf{y}}^T$, which is the argument of the $\log\det$ function above for $\varepsilon \to 0$.  As a result, we can view $h_{LD}^{(\varepsilon)}(\mathbf{x}|\mathbf{y})$, which is the LD-Entropy of $\mathbf{e}_{MMSE}$, as a measure of the remaining uncertainty after linearly (affinely) estimating $\mathbf{x}$ from $\mathbf{y}$ based on the MMSE criterion."

---

> ### Author Response · Authors · 2022-08-02
> **Reponse to Reviewer GZwK Part 1 of 4**
>
> We thank you for your time and detailed review. We appreciate that you found our framework effective and our article easy to read. In the revised article,  we have addressed your comment about extending the experimental results. In our response below and in the revised article, we have also tried to clarify the novelty of our framework.
>
> > The novel of this paper is limited. The main contribution of this paper is to directly generalize the earlier proposed log-determinant mutual information to the field of self-supervised learning.
>
> The application of information-theoretic metrics to improve machine learning in general and self-supervised learning in particular is an active research topic. What metric to use, which approximation to compute, and where to apply these metrics (e.g. between input output vs. between alternative outputs) are still being explored. We believe that our paper introduces multiple novel contributions, which should be of interest to further research in this area:
> - Based on information theoretical grounds, we propose a novel framework (CorInfoMax) with an interpretable loss function that has explicit components for minimizing variance of positive samples (the attraction factor) and for making full use of the embedding space, avoiding dimensional collapse (the big-bang factor).
> - We introduce a computationally-efficient approximation to LDMI that simplifies it by using an identity mapping instead of a general linear mapping.  This approximation directly minimizes representation variance, which is one of the objectives of self-supervised learning.
> - CorInfoMax relies only on second-order statistics, does not require negative samples, and does not force embeddings to be uncorrelated.
> - In addition to these theoretically appealing features, the proposed CorInfoMax framework achieves either better or competitive performance results relative to the state-of-the-art SSL approaches.
>
> In order to clarify these main contribution points, we included the list above in the introduction section of the revised article.
>
> > Section 3 is superfluous.
>
> Our goal with Section 3 was to provide a sufficient background on the log-determinant mutual information measure, which is instrumental in understanding the CorInfoMax loss proposed in the article. We specifically kept this section more detailed, as most NeurIPS readers might not have complete familiarity with the concepts. However, following your suggestion, we have shortened the discussion in the main text. We have moved the basic background about Shannon mutual information and LDMI to the Appendix. With this change, the main text contains only the discussion on the application of LDMI to the SSL problem and the derivation of the CorInfoMax loss from the LDMI measure.  Hopefully it now reads more concisely.
>
> > The experimental part is relatively weak. (i) This article only contains the results of classification tasks, and the performance of the proposed method on tasks such as detection, transfer, and semi-supervised learning also needs to be given. (ii) The classification results on ImageNet-1000K should be reported.
>
> In the revised article, we have expanded the experimental results part of our article, which addresses the concerns raised by the reviewer:
>
> - We have included 100-epoch linear classification and semi-supervised learning (fine-tuning)  results for Imagenet-1K in Section 5 of the article,
>
> - We have included transfer learning results in Appendix I of the article,
>
> - In Appendix K, we have included the eigenvalue plot for the projector covariance matrix after the CorInfoMax training to demonstrate that the proposed framework avoids dimensional collapse,
>
> - In Appendix L, we have provided a graph demonstrating the progress of Log-Determinant Mutual Information and online test accuracy during the CIFAR-10 CorInfoMax training, to demonstrate steady improvement of both values.
>
> - We have provided a hyperparameter sensitivity study in Appendix F.6.
>
> - We have also updated our accuracy results for  CIFAR, Tiny-Imagenet and Imagenet-100 datasets to report further enhancement in our algorithm's performance due to hyperparameter optimization.
>
> With these additions, we have a more comprehensive coverage of the experimental study of our framework.

---

### Official Review · Reviewer_J7ne · 2022-07-11

**Rating:** 6
**Confidence:** 4
**Soundness:** 4 excellent
**Presentation:** 3 good
**Contribution:** 3 good

**Summary:**

The paper introduced CorInfoMax, a second order statistics based mutual information method to avoid collapse problem in self supervised learning (SSL). It used log-determinant mutual information (LDMI) as the measure between data pairs with different augmentations. Mathematically, the paper gave step-by-step formulation to derivate the final form of the objective via its first-order Taylor series approximation
Experimental results on CIFAR10/CIFAR100/ImageNet gives competitive results.


**Questions:**

As commented in strengths and weakness

**Limitations:**

Yes

**Strengths And Weaknesses:**

Strengths
- This work is novel and well intuited. Collapse issue is a well-known problem in SSL, traditionally Shannon Mutual Information (SMI) is applied to mitigate the collapse issue with several drawbacks, such as large batch requirement. The log-determinant mutual information (LDMI) naturally enforces latent distributions have non-degenerate distributions which can avoid collapse. Therefore, this method is well motivated and makes sense
- Clean and neat formula. Step by step formula gives clean loss format in Eq (8). It clearly indicated data under different augmentation should be closed to each other and log determinant formula enforces the model from model collapse. The result is elegent.
- This result shows state-of-the-art SSL performance.

Weakness
- Mathematical formula is too heavy in the paper. It is a bit hard for audience to follow. I would strongly recommend a more detailed inference in any appendix to show how they are coming from
- More though experiments should be provided to demonstrate the effectiveness of the approach. For example, the paper argues the drawbacks for SMI and introduce LDMI measure as a better solution to avoid collapse. However, no evidence shown hot does it better avoids collapse.

---

> ### Author Response · Authors · 2022-08-02
> **Response to Reviewer J7ne**
>
> We thank you for your review and comments. We are glad that you found our framework novel, our formulation neat, and you evaluated our CorInfoMax framework as elegant.  In the following, we provide our response to your comments and suggestions.
>
> > Mathematical formula is too heavy in the paper. It is a bit hard for audience to follow. I would strongly recommend a more detailed inference in any appendix to show how they are coming from.
>
> Following your suggestion to improve the presentation, we have moved part of the mathematical details in Section 3 to Appendix A. We hope that the article is now more concise and has a better flow.
>
> > More though experiments should be provided to demonstrate the effectiveness of the approach. For example, the paper argues the drawbacks for SMI and introduce LDMI measure as a better solution to avoid collapse. However, no evidence shown hot does it better avoids collapse.
>
> Regarding experiments: We have added multiple experimental updates, including
>
>   - We have included 100-epoch linear classification and semi-supervised learning (fine-tuning)  results for Imagenet-1K in Section 5 of the article,
>
>   - We have included transfer learning results, in Appendix I of the article,
>
>   - In Appendix K, we have included the eigenvalue plot for the projector covariance matrix after the CorInfoMax training to demonstrate that the proposed framework avoids dimensional collapse,
>
>   - In Appendix L, we have provided a graph demonstrating the progress of Log-Determinant Mutual Information (LDMI) and online test accuracy during the CIFAR-10 CorInfoMax training, to demonstrate steady improvement of both values.
>
>   - We have provided a hyperparameter sensitivity study in Appendix F.6, which demonstrates that the CorInfoMax approach is fairly robust to hyperparameter changes.
>
>   - We have also updated our accuracy results for  CIFAR, Tiny-Imagenet and Imagenet-100 datasets to report further enhancement in our algorithm's performance due to hyperparameter optimization.
>
> Regarding Collapse:
>
> Theoretically, for $\varepsilon\to 0$, the $\log\det(\mathbf{\hat{R}}\_{\mathbf{z}^{(1)}}[l]+\varepsilon \mathbf{I})$ and $\log\det(\hat{\mathbf{R}}\_{\mathbf{z}^{(2)}}[l]+\varepsilon\mathbf{I})$ components of the CorInfoMax objective would become $-\infty$ in the event of dimensional collapse. Therefore, these components of the objective function  clearly avoid dimensional collapse, prohibiting restriction to any subspace. In other words, the $\log\det$ components of the CorInfoMax objective clearly protect against not only point collapse, but also dimensional collapse. To further clarify this point, we have added Appendix K to the revised article, where we provide an illustration for the eigenvalues of the projector covariance matrix $\hat{\mathbf{R}}\_{\mathbf{z}^{(1)}}[l]$ to show that the energy is spread in all dimensions of the embedding space, demonstrating that the CorInfoMax objective clearly prevents collapse.

---

### Official Review · Reviewer_nwQX · 2022-07-11

**Rating:** 6
**Confidence:** 4
**Soundness:** 3 good
**Presentation:** 3 good
**Contribution:** 3 good

**Summary:**

This paper extensively addresses the collapse problem by proposing second-order statistics-based mutual information measure that reflects the level of correlation among the inputs. It claims that maximizing this correlative information measure between alternative representations of the same input serves two purposes: (1) it avoids the collapse problem by generating feature vectors with non-degenerate covariances; (2) it establishes relevance among alternative representations by increasing the linear dependence among them. All these are proved to simplify as a regularization term which acts as a natural barrier against feature space degeneracy. In general, the paper is well written and reasonably understandable. I found the underlying theory to be very strong, however the presented experiments aren't sufficient to show the strength of the work.

**Questions:**

(1) Currently, the abstract is quite long and puts a lot of unnecessary details and background information, which fades out the necessary information. It does not mention anything on the achieved results.

(2) Space missing in line 201 between "linear" and "(affine)"

(3) In line 330, the phrase "objective function" appears twice, which sounds repetitive.

(4) In equation (10), the dimensionality of the projector is bottleneck because of the correlation matrix computation. Could you please comment on that?

(5) In table 1, under the CIFAR-10 and ImageNet-100/ResNet-18 column, the boldfaced numbers are mistakenly written?

(6) It would be good to direct the main comparison table in the supplementary in the main paper.

(7) Appropriate direction to different sections of the supplementary material should be done from the main paper.

(8) In figure 2, the drawing of encoder and projector could have been in different colours. I mean if they share weights, they get the same colour, otherwise if they don't share weights, they get different colours.

**Limitations:**

The authors have not mentioned about the limitation of the work in the paper. A possible limitation could be the consideration of only the global information and not considering the local information, which might make this method only applicable for global task, such as classification. I suspect this type of model won't work very well on the task like involving fine-grained feature understanding.


**Strengths And Weaknesses:**

**Strengths**

(1) The paper is theoretically grounded on correlative information measure of representation. Additionally, it is very well reasoned, I like the way it is presented.

(2) On classification downstream task, the proposed methods has obtained state-of-the-art results one some datasets. Although, I found that the results are not robust and the comparison is not complete. Please also see my experiment related comments in the weaknesses section.

**Weaknesses**

(1) The experiments shown are bit limited and not very robust. Currently the experiments are only done on one downstream task i.e. classification. More experiments on other tasks, such as object detection, semantic segmentation, few-shot learning, representation transferability etc could have been interesting and could have strengthen the method. Moreover, ablation of the model on different hyperparameters should have been studied to understand the strength of the model. Furthermore, standard experimental strategies are not followed, as an example, for classification tasks, self-supervised models follow "linear" and "fine-tuning" strategies, which is not followed in this work.

(2) I wonder if there is any way to estimate the amount of mutual information maximized via the proposed CorInfoMax approach. It would be interesting to see some intermediate results or plots showing the evolution of estimated mutual information as the training progresses. Also it would have been interesting to see analysis on the performance behaviour of the model with evolution of mutual information. Does higher mutual information always guarantee a better trained model?

---

> ### Author Response · Authors · 2022-08-02
> **Response to Reviewer nwQX Part 2 of 2**
>
> > In table 1, under the CIFAR-10 and ImageNet-100/ResNet-18 column, the boldfaced numbers are mistakenly written?
>
> Thanks for this comment. As written in the caption of the table, we bold all top results that are statistically indistinguishable, which is determined by the level of accuracy and the number of instances. In 5.2 Results section, we also state "Due to the limited size of
> the validation sets (5-10K), differences less than ≈ 0.5$\%$ are not statistically significant."
>
> > It would be good to direct the main comparison table in the supplementary in the main paper.
>
> Thanks for this suggestion. In Section 5.2 of the article, we have included a reference to the full comparison table in Appendix H.
>
> > Appropriate direction to different sections of the supplementary material should be done from the main paper.
>
> We have placed references to all sections of the Appendix in the appropriate parts of the article.
>
> > In figure 2, the drawing of encoder and projector could have been in different colours. I mean if they share weights, they get the same colour, otherwise if they don't share weights, they get different colours.
>
> Thank you for your suggestion. We use the blue color code for "mapping" blocks consistently (either networks or augmentations).  To distinguish the weight-shared case, we have added a separate figure  (Figure A.1) provided in  Appendix B.
>
> > Limitations: The authors have not mentioned about the limitation of the work in the paper. A possible limitation could be the consideration of only the global information and not considering the local information, which might make this method only applicable for global task, such as classification. I suspect this type of model won't work very well on the task like involving fine-grained feature understanding.
>
> In Appendix I of the revised article, we report that the CorInfoMax achieves competitive performance for object detection and segmentation relative to the state of the art in SSL. We look at further improvements related to fine-grained feature understanding as future extensions of our framework. As a limitation, we have mentioned the difficulty in selecting the correct augmentations and hyper-parameters in the discussion section.

---

> > ### Comment · Reviewer_nwQX · 2022-08-08
> > **Response to rebuttal**
> >
> > I appreciate the efforts the authors have put in the rebuttal. They have added explanations, clarifications and new experiments, which I believe to improve the overall understandability and the main contribution of the paper. In the light of the new explanations and experiments offered to me and other reviewers, I have more confidence in the correctness and motivations for their work. I believe the updated paper together with the appendix and supplementary material should contribute to knowledge for the community. I have updated my rating for the paper.

---

> > > ### Author Response · Authors · 2022-08-09
> > > **Thank you**
> > >
> > > We would like to thank you for your useful feedback and constructive suggestions.

---

> ### Author Response · Authors · 2022-08-02
> **Response to Reviewer nwQX Part 1 of 2**
>
> We thank you for your comments and suggestions. We appreciate your positive comment: "...the paper is well written and reasonably understandable. I found the underlying theory to be very strong". We revised our article to address your main concern about the experimental part. We provide below our response to your comments.
>
> > The experiments shown are bit limited and not very robust. Currently the experiments are only done on one downstream task i.e. classification. More experiments on other tasks, such as object detection, semantic segmentation, few-shot learning, representation transferability etc could have been interesting and could have strengthen the method. Moreover, ablation of the model on different hyperparameters should have been studied to understand the strength of the model. Furthermore, standard experimental strategies are not followed, as an example, for classification tasks, self-supervised models follow "linear" and "fine-tuning" strategies, which is not followed in this work.
>
> In the revised article, we have addressed the concerns you listed about the experimental part. In particular,
> - We have included linear classification and semi-supervised learning (fine-tuning)  results for Imagenet-1K in Section 5 of the article,
> - We have included transfer learning results for object segmentation and detection task in Appendix I of the article,
> - In Appendix K, we have included the eigenvalue plot for the projector covariance matrix after the CorInfoMax training to demonstrate that the proposed framework avoids dimensional collapse,
> - In Appendix L, we have provided a graph demonstrating the progress of Log-Determinant Mutual Information and online test accuracy during the CIFAR-10 CorInfoMax training, to demonstrate steady improvement of both values.
> - We have provided a hyperparameter sensitivity study in Appendix F.6, which demonstrates that the CorInfoMax approach is fairly robust to hyperparameter changes.
>
> > I wonder if there is any way to estimate the amount of mutual information maximized via the proposed CorInfoMax approach. It would be interesting to see some intermediate results or plots showing the evolution of estimated mutual information as the training progresses. Also it would have been interesting to see analysis on the performance behaviour of the model with evolution of mutual information. Does higher mutual information always guarantee a better trained model?
>
> We note that CorInfoMax cost is a variation on the LDMI cost (with an addition constraint on the linear mapping). Therefore, it is natural to expect that LDMI increases as the model converges. In order to clarify his point, in  Appendix L of the revised article, we illustrate the evolution of the LDMI measure, which is an estimator of SMI under Gaussian assumption, and the linear classification accuracy during the CorInfoMax objective optimization. These plots demonstrate the general case where both the LDMI and the testing accuracy together increase in a correlated manner. This confirms the suitability of LDMI (and CorInfoMax) as a sensible measure for the SSL criterion.
>
> > Currently, the abstract is quite long and puts a lot of unnecessary details and background information, which fades out the necessary information. It does not mention anything on the achieved results.
>
> Thank you for your suggestion. We have modified the abstract by reducing its length and reflecting our experimental performance results.
>
> > Space missing in line 201 between "linear" and "(affine)"
>
> Corrected. Thanks a lot.
>
> > In line 330, the phrase "objective function" appears twice, which sounds repetitive.
>
> Removed the first one. Thanks a lot.
>
> > In equation (10), the dimensionality of the projector is bottleneck because of the correlation matrix computation. Could you please comment on that?
>
> Section 4.3 (Computational Complexity) and Appendix G (Algorithm runtime results) address the issue of the impact of $\log\det$ covariance term on computational complexity. We have further extended this discussion to include runtime results for the Imagenet-1K dataset. These results are reported in Table 9 of Appendix G. From these results, we observe that the impact of the $\log\det$ function on computational complexity is negligible compared to computational complexity due to the other parts of model training. This is mainly due to the fact that the encoder complexity scales proportionally to the input dimension, whereas the complexity of $\log\det$ is dependent only on the projector dimension. In fact, as demonstrated in Table 9 of Appendix G, for the Imagenet-1K dataset and for a projector dimension of $1024$ and for a batch size of $512$, the computation $\log\det$ contributes only $2.83$ percent of the total computation time, which is negligible. This percentage also drops with increasing batch sizes.

---

### Official Review · Reviewer_sfru · 2022-07-12

**Rating:** 6
**Confidence:** 4
**Soundness:** 3 good
**Presentation:** 3 good
**Contribution:** 3 good

**Summary:**

This paper presents a self-supervised learning with an information maximization criterion among alternative latent representations of the same input that naturally prevents dimensional collapse. It considers a second-order-statistics based mutual information measure, the log-determinant mutual information (LDMI), which is equivalent to Shannon mutual information under Gaussian distribution. A further first-order approximation to the log-determinant of the sum of two matrices is used to simplify the final objective to a Euclidean distance-based objective function regularized by the log-determinant of the feature covariance matrix. Consequently it  avoids the collapse problem establishes relevance among alternative representations by increasing the linear dependence among them.  Experiments on 4 image datasets show that the proposed approach gives better results than contrastive and non-contrastive methods.

**Questions:**

The proposed method maximizes Shannon mutual information under Gaussian distribution, and a further first-oder approximation is also used.

Experiments on synthetic Gaussian data set in (Poole et al. 2019, McAllester and Stratos 2020) are need to see the effect of first-oder approximation when estimating mutual information, and the result should compare with the DoE method in McAllester and Stratos' paper.

Finally are there any theoretical explanations to explain why it gets better results than contrastive and non-contrastive methods? The contrastive methods are essentially a lower bound method of Shannon mutual information.

**Ethics Review Area:**

["I don’t know"]

**Limitations:**

Yes

**Strengths And Weaknesses:**

The paper is well written and easy to follow. However, there are several issues I'd like to address. The revised version has cleared my concerns.

1. This following statement is misleading: "A common self-supervision task is to match the latent representations that come from the distortions of the same input." It's true in computer vision, but not true in speech or natural language processing.

2. The following statement is not quite right: "maximizing SMI between the representations of the same input is a challenging task whose implementation would require relatively large sample sizes [11, 12]." [11] demonstrates that when mutual information is large,  existing variational lower bounds degrade and exhibits either high bias or high variance. In fact McAllester and Stratos prove that serious statistical limitations are inherent to any lower bound method of measuring mutual information. More specifically, any distribution-free high-confidence lower bound on mutual information estimated from N samples cannot be larger than O(ln N).

3. Collapse is a central concern for SSL with the same input in computer vision.

Reference:
David McAllester and Karl Stratos, Formal Limitations on the Measurement of Mutual Information, The Twenty Third International Conference on Artificial Intelligence and Statistics, 108:875-884, 2020.

---

> ### Author Response · Authors · 2022-08-02
> **Response to Reviewer sfru**
>
> We would like to thank you for reviewing our paper, please find our detailed responses below.
>
> > This following statement is misleading: "A common self-supervision task is to match the latent representations that come from the distortions of the same input." It's true in computer vision, but not true in speech or natural language processing.
>
> Thank you for pointing this out. We modified the related sentences to point out that this is a computer vision related pretext task.
>
> > The following statement is not quite right: "maximizing SMI between the representations of the same input is a challenging task whose implementation would require relatively large sample sizes [11, 12]." [11] demonstrates that when mutual information is large, existing variational lower bounds degrade and exhibits either high bias or high variance. In fact McAllester and Stratos prove that serious statistical limitations are inherent to any lower bound method of measuring mutual information. More specifically, any distribution-free high-confidence lower bound on mutual information estimated from N samples cannot be larger than O(ln N).
>
> We appreciate that you have brought this point and the relevant reference (the McAllester-Stratos article) to our attention. We have added this reference and corresponding clarifying claims in our revised article, which particularly improved our discussion about the SMI based methods. We have modified the sentence you pointed out as "For SSL, the fundamental limitations on the precise computation of SMI [11,12,13] impose serious challenges on its optimization. Especially when mutual information is large, we require drastically more samples (exponentially related to SMI) to obtain a reliable estimate of SMI [13]."
>
> > Collapse is a central concern for SSL with the same input in computer vision.
>
> Thank you.  We have corrected this sentence as "Collapse is a central concern for SSL with the same input in computer vision".
>
> > The proposed method maximizes Shannon mutual information under Gaussian distribution, and a further first-oder approximation is also used.  Experiments on synthetic Gaussian data set in (Poole et al. 2019, McAllester and Stratos 2020) are need to see the effect of first-oder approximation when estimating mutual information, and the result should compare with the DoE method in McAllester and Stratos' paper.
>
> The precision of LDMI variations as a Shannon Mutual Information approximator is an interesting topic in its own right and may be the subject of future work. However, we would like to clarify that our goal in this article is not to obtain a precise and efficient approximation of Shannon Mutual Information (SMI), which is the main focus of the McAllester and Stratos article. Instead, we directly propose to work with LDMI to derive a linear dependence measure and place an additional (identity) constraint on the corresponding linear mapping. Therefore, we believe that the proposed comparison may detract from the focus of the article, as the CorInfoMax objective is not intended as an SMI approximator but derived as a restricted linear dependence measure for SSL. In the process of obtaining the CorInfoMax objective function, we perform a first-order approximation on some of the $\log\det$ terms in (1) and, in addition, we enforce the identity mapping, instead of the inherent MMSE linear estimator mapping, as shown in (2). Therefore, the expression in (2) does not function as an SMI estimator. Furthermore, comparing the performance of the LDMI expression in (1), which would be the SMI expression in the Gaussian setting, with an SMI estimator without a distribution prior for the synthetic Gaussian data setting would not be relevant. However, it could be possible that the methods in these articles can be helpful in deriving alternative objectives based on the LDMI measure, potentially improving computational complexity, which we leave to future work.
>
> > Finally are there any theoretical explanations to explain why it gets better results than contrastive and non-contrastive methods? The contrastive methods are essentially a lower bound method of Shannon mutual information.
>
> As we argue in the article, a deterministic but nonlinear relationship between two representations would maximize SMI; however, it would not necessarily be a favorable organization of the embedding space for later classification, e.g.,  with a linear separation operation. The special linear dependence form imposed on positive samples by CorInfoMax reduces the representation variance due to random distortions. At the same time, the $\log\det$ function, aka "the big-bang factor", conveniently spreads embeddings across the feature space. Formal analysis of how the optimization of the CorInfoMax objective impacts downstream tasks is on our future research agenda.

---

### Author Response · Authors · 2022-08-02
**Authors' General Response to Reviewer Comments**

We would like to thank all reviewers for their time and valuable feedback. In the following, we provide our response to their comments and our answers to their questions. We made changes to the article and the supplementary document based on the comments we received from the reviewers.

Summary of the main changes:

* We have updated the abstract by shortening and referencing our results,

* We have modified the introduction to clarify our novelty/contributions,

* We have moved part of the theoretical background on information measures to Appendix A to improve the flow in the main text,

* We have added multiple experimental updates, including

  - We have included 100-epoch linear classification and semi-supervised learning (fine-tuning)  results for Imagenet-1K in Section 5 of the article,

  - We have included transfer learning results, in Appendix I of the article,

  - In Appendix K, we have included the eigenvalue plot for the projector covariance matrix after the CorInfoMax training to demonstrate that the proposed framework avoids dimensional collapse,

  - In Appendix L, we have provided a graph demonstrating the progress of Log-Determinant Mutual Information (LDMI) and online test accuracy during the CIFAR-10 CorInfoMax training, to demonstrate steady improvement of both values.

  - We have provided a hyperparameter sensitivity study in Appendix F.6, which demonstrates that the CorInfoMax approach is fairly robust to hyperparameter changes.

  - We have also updated our accuracy results for  CIFAR, Tiny-Imagenet and Imagenet-100 datasets to report further enhancement in our algorithm's performance due to hyperparameter optimization.

---

### Author Response · Authors · 2022-08-08
**An update on Imagenet-1000 linear evaluation and semi-supervised learning (fine tuning) accuracy results**

Dear Reviewers,

This is a brief update about our Imagenet-1000 results: With hyperparameter optimizations, we improved CorInfoMax’s linear evaluation Top-1 accuracy to $69.08$  (previous value: $68.56$) and semi-supervised (fine tuning) accuracy to  $44.89$  for %1 of samples and $64.36$ for %10 of samples  ( previous values:  $44.35$  for %1 of samples and $63.81$ for %10 of samples).

We updated our article (especially Table 1 and Table 2)  and reviewer responses accordingly.

We would like to thank all reviewers for their efforts.

---

### Meta-Review · Area_Chair_1ac8 · 2022-08-23

**Recommendation:** Accept
**Confidence:** Certain

**Metareview:**

The paper describes a self-supervised learning method based on an information maximization criterion that naturally prevents dimensional collapse. The authors consider the Shannon mutual information under the assumption that the data is Gaussian. A first-order approximation to the log-determinant of the sum of two matrices is used to simplify the final objective. Experiments on 4 image datasets show that the proposed approach gives better results than contrastive and non-contrastive methods.

Strengths:

1 - The paper is well written and easy to follow.
2 - The paper is theoretically grounded on correlative information measure of representation.
3 - Strong results on some downstream classification problems.
4 - Initially the experiments included only one downstream task regarding classification, but the paper has been updated to include also results for object segmentation and detection task.
5 - Novel and well motivated.
6 - state-of-the-art SSL performance.

Weaknesses:

- Some weaknesses are pointed out by reviewer GZwK, but these are not well justified.

Decision:

A majority of reviewers vote for acceptance. The only reviewer voting slightly towards rejection is GZwK, with a reasoning that is not well justified. For example, the main criticisms mentioned by reviewer GZwK

- The paper directly generalizes the earlier proposed log-determinant mutual information to the field of self-supervised learning.
- this paper does not give a deep-going analysis that why the second-order statistics can play a important role in self-supervised learning

are not mentioned by any of the other reviewers.

Because of this, I have decided to accept the paper.



**Award:**

No

---

### Decision · Program_Chairs · 2022-09-14

Accept